# Construction of sublingual trilaminated Eszopiclone fast dissolving film for the treatment of Insomnia: Formulation, characterization and In vivo clinical comparative pharmacokinetic study in healthy human subjects

**Mahmoud Teaima**[1]*, **Mohamed Yasser**[2], **Nehal Elfar**[2], **Kamel Shoueir**[3], **Mohamed El-Nabarawi**[1], **Doaa Helal**[4]

1 Department of Pharmaceutics and Industrial Pharmacy, Faculty of Pharmacy, Cairo University, Cairo, Egypt, 2 Department of Pharmaceutics and Industrial Pharmacy, Faculty of Pharmacy, Horus University, New Damietta, Egypt, 3 Institute of Nanoscience & Nanotechnology, Kafrelsheikh University, Kafrelsheikh, Egypt, 4 Department of Pharmaceutics, Faculty of Pharmacy, Fayoum University, Fayoum, Egypt

* mahmoud.teaima@pharma.cu.edu.eg

## Abstract

### Background

Disturbed sleep can cause to m health problems such as cognitive impairment, depressed mood, and negative effects on cardiovascular, endocrine, and immune function. This study formulates and optimizes Eszopiclone trilaminate fast dissolving film.

### Methods

Prepared Eszopiclone trilaminate fast dissolving film (Eszopiclone TFDF) was characterized by disintegration time, drug release, tensile strength (TS), percentage elongation (EB%), folding endurance, taste masking test, and *in vitro* dissolution test. The selected formulas were F2 (0.5% xanthan gum, 10% propylene glycol), F4 (3% sodium alginate, 10% propylene glycol) and F6 (1.5% pullulan, 10% propylene glycol) were subjected to *in vivo* study compared to conventional Lunesta® tablet.

### Results

The results indicated that disintegration time was in the range of 940 m. Drug release was found to be in the field of 78.51%–99.99%, while TS values and EB% differed from 11.12 to 25.74 (MPa) and 25.38%–36.43%, respectively. The folding endurance went between 200 and 300 times. All formulas exhibited acceptable uniformity content, surface **pH**, film thickness, and a good taste feeling.

**Data Availability Statement:** All relevant data are available publicly at the following link: https://figshare.com/s/2bf90b42643556628619.

**Funding:** The author(s) received no specific funding for this work.

**Competing interests:** The authors have declared that no competing interests exist.

## Conclusion

F4 had the highest $C_{max}$ (39.741 ± 6.785-μg/l) and lower $T_{max}$ (1.063 hr) among other formulas and conventional tablets. Therefore, FDFs' technology could increase the therapeutic effect of Eszopiclone.

## 1. Introduction

Insomnia can cause many health problems, such as cognitive impairment, depressed mood, and adverse effects on cardiovascular, endocrine, and immune function [1]. In the US, Eszopiclone is the first hypnotic agent that does not restrict its length of use. It can be used for 6–12 months without evidence of problems (e.g., tolerance/dependence) [2]. The FDA approved Eszopiclone (Lunesta®) on December 15th, 2004.

Eszopiclone ($C_{17}H_{17}ClN_6O_3$) is a non-benzodiazepine hypnotic agent of the cyclopyrrone class (hypnotic sedative class of drugs) with a molecular weight of 388.81. It is a white to light-yellow crystalline solid, slightly soluble in water, slightly soluble in ethanol, and soluble in phosphate buffer (**pH** 3.2) [3, 4]. Eszopiclone is rapidly absorbed following oral administration. Its peak plasma concentrations are achieved about an hour after oral administration. Eszopiclone is weakly bound to plasma protein (52%–59%). The most dramatic side effect of Eszopiclone is its unpleasant taste in the mouth, so it has been formulated as film-coated tablets to be used for oral administration in 1-mg, 2-mg, or 3-mg tablets [4].

Fast dissolving films (FDF) have shown the ability to mask the drug's bitter taste, enhance the onset of drug action, decrease the dose frequency, convenient dosing, and enhance the drug efficacy [5, 6]. Dose accuracy in comparison to syrup is also advantage of fast dissolving film [7, 8]. The disadvantage of FDF is that high doses cannot be incorporated into the strip. The amount should be between 1–40-mg. There remain many technical limitations with film strip use; the thickness while casting the film. Glass petri- plates cannot be used for casting. The other technical problem with these dosage forms is achieving dose uniformity. Packaging of films needs special equipment, and it is not easy to pack [6, 9, 10].

An ideal fast dissolving delivery system should have the following qualities: high stability, transportability, ease of handling and administration, no special packaging material or processing requirements, no water necessary for application[6, 11].

The formulation of Eszopiclone as a fast-dissolving film may lead to masking the bitter drug taste, very rapid absorption with a fast onset of action, and can also enhance its bioavailability.

This study formulates and optimizes fast dissolving film containing Eszopiclone in addition to studying the influences of formulation parameters on film attributes (*in vitro* disintegration time, drug content, percent elongation, tensile strength (TS), folding endurance, taste masking test, *in vitro* dissolution test, and *in vivo* bioavailability).

## 2. Material and methods

### 2.1. Chemicals and excipients

Eszopiclone was bought from the Western company, Egypt. Pullulan, Hydroxypropyl methylcellulose (HPMC) were gifts from EVA Pharma, Egypt. Potassium dihydrogen orthophosphate was bought from Winlab (Leicestershire, United Kingdom). Sodium phosphate dibasic was supplied from Sigma-Aldrich (Missouri, USA). Glycerol, propylene glycol, gelatin, sodium

alginate, xanthan gum, clove oil, peppermint oil, and sucralose were bought from Al-Gomhoria Company for medicines and medicals, Cairo, Egypt. All other reagents and solvents used were of analytical grade.

## 2.2. Methodology

**2.2.1. Differential scanning calorimetry (DSC).** DSC test was used to identify the melting temperature and material purity or interaction using a differential scanning calorimeter (Shimadzu DSC TA-50 ESI, Tokyo, Japan). The incompatibilities were detected by appearance, shift, or disappearance of the corresponding peaks.

**2.2.2. Construction of Eszopiclone triple fast dissolving films.** Taste masking techniques and triple layer films are realistic to overcome the very bitter taste of Eszopiclone as follows.

**2.2.3. Preparation of medicated fast dissolving films containing Eszopiclone.** FDF were prepared using the solvent casting technique [12]. Polymers (Gelatin, xanthan gum, sodium alginate, and pullulan) and propylene glycol were used as a plasticizer with different concentrations (5%–10%) chosen from the preliminary study to form FDF containing the calculated amount of drug (Eszopiclone). The composition of different formulas is provided in Table 1.

**2.2.4. Preparation of plain film for sweetening effect and plain film for flavoring effect.** Specified weight of film-forming polymer HPMC (1.5%) was first dissolved in 30-mL casting solvent (distilled water) while stirring on a magnetic stirrer (L32; Bibby, Staffordshire, UK). the polymeric solution was divided into two equal portions. Sweetener (Sucralose) was dissolved in the first portion of the polymeric solution, and the calculated amount of plasticizer propylene glycol (10%) was added to form plain film for sweetening effect. Clove oil, peppermint oil (flavoring agents) was added to the second portion of the polymeric solution to give pain relief and the mouth refreshment feeling. The final volume of each was adjusted to 20-mL with a solution containing (distilled water: isopropyl alcohol (1:1)). And the beaker was covered with aluminum foil to prevent solvent evaporation. The casting solution was subjected to gentle stirring for two hours using a magnetic stirrer (L32; Bibby, Staffordshire, UK).

The casting solution (20-mL) was transferred into a previously cleaned and dried petriplate (Glass petri- dishes; Desiccators, glads funnels (diameter was 9-cm)). Then, the solution was cast on petri-dish and left to evaporate at room temperature for the next 24 h. After drying, the film was removed from the petri-plate and cut into the desired size (1-cm x 1-cm). Then, wrapped in an aluminum foil (to maintain the integrity and elasticity of the films) and were finally stored in a dry place at ambient room temperature.

**2.2.5. Formation of trilaminated FDFS (sandwich appearance).** The film containing the flavoring agent was placed on the surface of the film containing the drug after spraying on the drug loaded FDF with small drops of casting solvent (isopropyl alcohol (1): water (4)) w/w on the surface of the film, the film containing the sweetening agent is placed on the other side of the film containing the drug, and the films stick together using a weight and left overnight. The films were subjected to evaluation within one week of preparation.

**2.2.6. Physicochemical In vitro evaluation of fast dissolving films (FDF).**

a. Visual examination: The smoothness, color, clarity, and transparency of oral films were examined visually. A subjective scoring system was applied to evaluate the prepared films. The score (+++) indicated complete peeling, flexibility, and transparency from the substrate. Films that could not be peeled from the substrate, even though they were transparent, took the score (++), fissured or very brittle films took (+).

**Table 1. Composition of medicated fast dissolving film containing eszopiclone.**

| Formula | Xanthan gum %w/w | Sodium alginate %w/w | Pullulan %w/w | Gelatin %w/w | Propylene glycol %w/w |
|---|---|---|---|---|---|
| F1 | 0.5% | - | - | - | 5% |
| F2 | 0.5% | - | - | - | 10% |
| F3 | - | 3% | - | - | 5% |
| F4 | - | 3% | - | - | 10% |
| F5 | - | - | 1.5% | - | 5% |
| F6 | - | - | 1.5% | - | 10% |
| F7 | - | - | - | 2.5% | 5% |
| F8 | - | - | - | 2.5% | 10% |

a. All formulas containing 63.585-mg Eszopiclone (1-mg in 1-cm$^2$ film).

b. The final volume was completed to 20-gm in all formulas with isopropyl alcohol: water (1:4) w/w.

c. All medicated film surrounded by a sweetening film and flavoring film explained below.

b.  Film thickness: Formulations were evaluated using a micrometer screw gage (Mitutoyo, Kawasaki, Japan) with an accuracy of 1-μm. Each sample film was measured using a micrometer at three places, and the calculated mean values [13].

c.  The **pH** of the surface: The **pH** was noted by placing the film in a petri-dish, slightly wet with the help of distilled water, then bringing the film's surface in contact with **pH** meter electrode (Mettler Toledo—Greifensee, Switzerland) for one minute [14].

d.  Drug content: Eszopiclone-loaded FDF was placed in a 100-ml volumetric flask, then dissolved in 50-ml phosphate buffer (**pH** 6.8) using a magnetic stirrer. The volume was completed using phosphate buffer (**pH** 6.8). The resulting solution was filtered to remove undissolved residue. One milliliter of the solution was further diluted to 10-ml with phosphate buffer solution (**pH** 6.8), and the absorbance was measured at 304-nm using a UV spectrophotometer (Biochrom Libra S22, Biochrom Ltd, Cambridge England) [15]. Drug content uniformity was conducted in triplicate [16]. A mean of three readings and standard deviation was recorded. Three films of each formula were used in this test [17].

**2.2.7. Mechanical In vitro evaluation of fast dissolving films (FDF) as tensile strength (TS), percentage elongation (EB%), and Folding endurance.**

a.  Tensile Strength Measurement (TS) is the maximum stress applied to a point at which the film specimen breaks and can be calculated by the following equation: Tensile Strength = Force at break (N)/ Cross-sectional area (mm$^2$) [17].
    Tensile testing was conducted using a texture analyzer AG/MC1 (Acquati, Italy) [18], equipped with a 5 N load cell. The film was cut into 30 × 20-mm strips. Tensile tests were conducted following the ASTM International Test Method for Thin Plastic Sheeting (D 882–02) [19]. Each test strip was placed in tensile grips on the texture analyzer. The initial grip separation was 20-mm, and crosshead speed was 1-inch/min. The maximum fracture force, i.e., the force attained just before the film strips ruptured, was recorded [20]. The measurements were repeated in triplicate using three film samples for each formulation type.

b.  Percentage elongation: When stress is applied, a strip sample stretches, and this is referred to as a strain. Strain is the deformation of the strip divided by the original dimension of the

sample. Generally, elongation of the strip increases as the plasticizer content increases [21].

$$elongation\ at\ break = (difference\ in\ length\ at\ breaking\ point \div original\ length) \times 100\%$$

c. Folding endurance (FE): is the most useful index of bending the flexibility of FDFs. It is measured by repeatedly folding the film at the same place until it breaks. The number of times the film could be folded at the same place until it broke or folded up to 300 times, which is considered satisfactory to reveal good film properties defined as FE [22, 23]. Three films of each formula were used in this test [24, 25].

**2.2.8. In vitro disintegration time (DT) of fast dissolving films (FDF).** DT was obtained by placing the film (1-cm$^2$) of each formula in a petri-plate in a beaker containing about 25-ml phosphate buffer (**pH** 6.8), and the time taken by each film to completely disintegrate was noted and considered as DT [16, 26]. Three films of each formula were tested.

**2.2.9. In vitro dissolution test of fast dissolving films (FDF).** *In vitro* dissolution tests of Eszopiclone-loaded oral films were performed using USP dissolution apparatus (Type II) (apparatus Pharma Test Apparatebau PT-DT70—Hainburg, Germany). The test was conducted at 37˚C ± 0.5˚C with stirring speed of 100-rpm ± 2 rpm in 100-ml phosphate buffer (**pH** 6.8). Samples were withdrawn at predetermined time intervals (1, 2, 3, 5, and 7 min) and replaced with the same volume of fresh buffer, in which sink conditions were maintained during dissolution. The absorbance was determined at $\lambda_{max}$ 304-nm using UV–visible spectrophotometer (Biochrom Libra S22, Biochrom Ltd Cambridge England) against a blank made of non-medicated films at the same conditions. The amount of drugs dissolved from each film was calculated. Three films of each formula were tested [27, 28].

**2.2.10. Taste masking test (human panel testing) of fast dissolving films (FDF).** Sensory evaluation of the taste of formulations was performed by six healthy volunteers [29] (three female and three male) aged (27–40) years are asked to take the bitter drug, and then the optimum taste-masked formulations selected to *in vivo* evaluation. Volunteers are then asked to comparatively rate the formulation on different organoleptic properties [30, 31]. A numerical scale was used with the following values: 0 –very pleasant, 1-pleasant, 2 –slightly pleasant, and 3-unpleasant.

**2.2.11. In vivo pharmacokinetic studies of fast dissolving films (FDF) in human volunteers.**

a. Experimental Design and Sample Collection: An *in vivo* study was conducted to compare the pharmacokinetic parameters of Eszopiclone from the optimized films (test products) and the commercially available, Lunesta® tablet containing 1-mg (reference product), after single oral administration. The protocol and informed consent form were approved by REC-FOPCU (PI 2313). Eight male healthy human volunteers were involved in the study. The study was fully explained to the volunteers before starting the study, and each volunteer signed a written informed consent form as per declaration of Helsinki (Brazil 2013), the study did not include minors, each subject signed the informed consent form by himself. Two treatments, four periods, randomized, and cross-over design were performed within a 1-week washout period as indicated in Fig 3A. The volunteers were randomly allocated to one of the two groups of equal size. They were only allowed to take water after one hour, and a standard meal was given after four hours from the administration of treatments. The study was supervised by a physician who was also responsible for the volunteers' safety and sample collection. Samples of venous blood (5-mL) were collected in heparinized blood tubes through the indwelling cannula immediately before oral dosing and at the predetermined time intervals of 0.25, 0.5, 0.75, 1, 1.5, 2, 3, 4, 5, 6, 7, 8, 9, 10, and 24 h after dosing.

The plasma samples were obtained by centrifuging the samples at 3500 rpm for ten minutes, and this was frozen at −20˚C in labeled tubes until further analysis.

b.  Sample preparation: About 50-μL Aripirazole (from a stock solution of concentration 20-ng/mL) was added to each sample (450-μL plasma) as an internal standard. Eszopiclone and Aripirazole were obtained using ethyl acetate (5-mL), and vortexed for 20 s, then centrifuged for one minute at 5000 rpm (cooling centrifuge, TGL-20 MB). The supernatant was transferred to other vials filtered through a 0.22-l-m Millipore filter, then evaporated to dryness using a vacuum concentrator (Eppendorf Vacufuge plus, Germany).

c.  LC-MS/MS Assay of Eszopiclone: Fig 3B represents a chromatogram of a confirmed LC-MS/MS method for analyzing plasma Eszopiclone concentrations, which was employed using LC-MS/MS system (Shimadzu®, Japan) coupled with a triple quadrupole detector (API-4500, AB Sciex, Foster, CA, USA). The mobile phase comprises Methanol: 0.2% Formic acid in water. The chromatographic separation was performed on the Sunfire C18 column (4.6 × 50-mm i.d., 5-μm diameter; Agilent, CA, USA). The injection volume was 10-μL, and the flow was isocratic at a rate of 0.2-mL/min. The method has been validated in terms of selectivity, linearity, precision, accuracy, carryover, extraction recovery, and stability.

d.  Statistical analysis of the Pharmacokinetic results: A non-compartmental pharmacokinetic model was applied to analyze the pharmacokinetic parameters of Eszopiclone after oral administration of two treatments using the PK solver program [32]. These parameters included peak plasma Eszopiclone concentration ($C_{max}$, ng/mL) and time to reach ($T_{max}$, h.), elimination rate constant (Kel, $h^{-1}$), terminal half-life ($t_{1/2}$, h.), area under the plasma concentration-time curve from time zero to the last observation time point ($AUC_{(0-24)}$, ng h/mL) and infinity ($AUC_{(0-\infty)}$, ng h/mL). IBM SPSS Statistics 20 (Armonk, NY, USA) were employed to analyze all statistical differences in data using a one-way ANOVA test for the established pharmacokinetic parameters, and $p$-value $< 0.05$ was statistically significant. Nonparametric Kruskal–Wallis test was conducted to compare the $T_{max}$ data obtained from two treatments.

## 3. Results and discussion

### 3.1. Differential scanning calorimetry (DSC)

The DSC thermogram of pure Eszopiclone has one main prominent sharp endothermic melting peak at about 206.85˚C (onset to 193.24˚C and endset to 218.75). The thermogram of xanthan gum was characterized by two thermal events: the first endothermic centered at about 104.01˚C, attributed to water loss associated with the hydrophilic groups of the polymer, and the second exothermic centered at about 290.28˚C, corresponding to the thermal degradation of xanthan. Three thermal events characterized sodium alginate: two are endothermic at about (93.41˚C and 248.91˚C), and the third has exothermically centered at about 290.6˚C. Gelatin was characterized by two endothermic thermal events: 81.52˚C and 216.41˚C. Meanwhile, broad peaks were observed at 67.14˚C, and 3118.19˚C from the DSC thermogram pullulan. The sharp endothermic peak of Eszopiclone was unchanged in the thermogram, confirming the absence of interaction between the drug and excipients used in the formulation. As indicated in Fig 1.

### 3.2. Physicochemical In vitro evaluation of fast dissolving films (FDF)

**3.2.1. Visual examination.**    Only films with a score (+++) were conducted for the *in vivo* evaluation. The formulas that shows the best appearance were F2, F4, and F6, which were

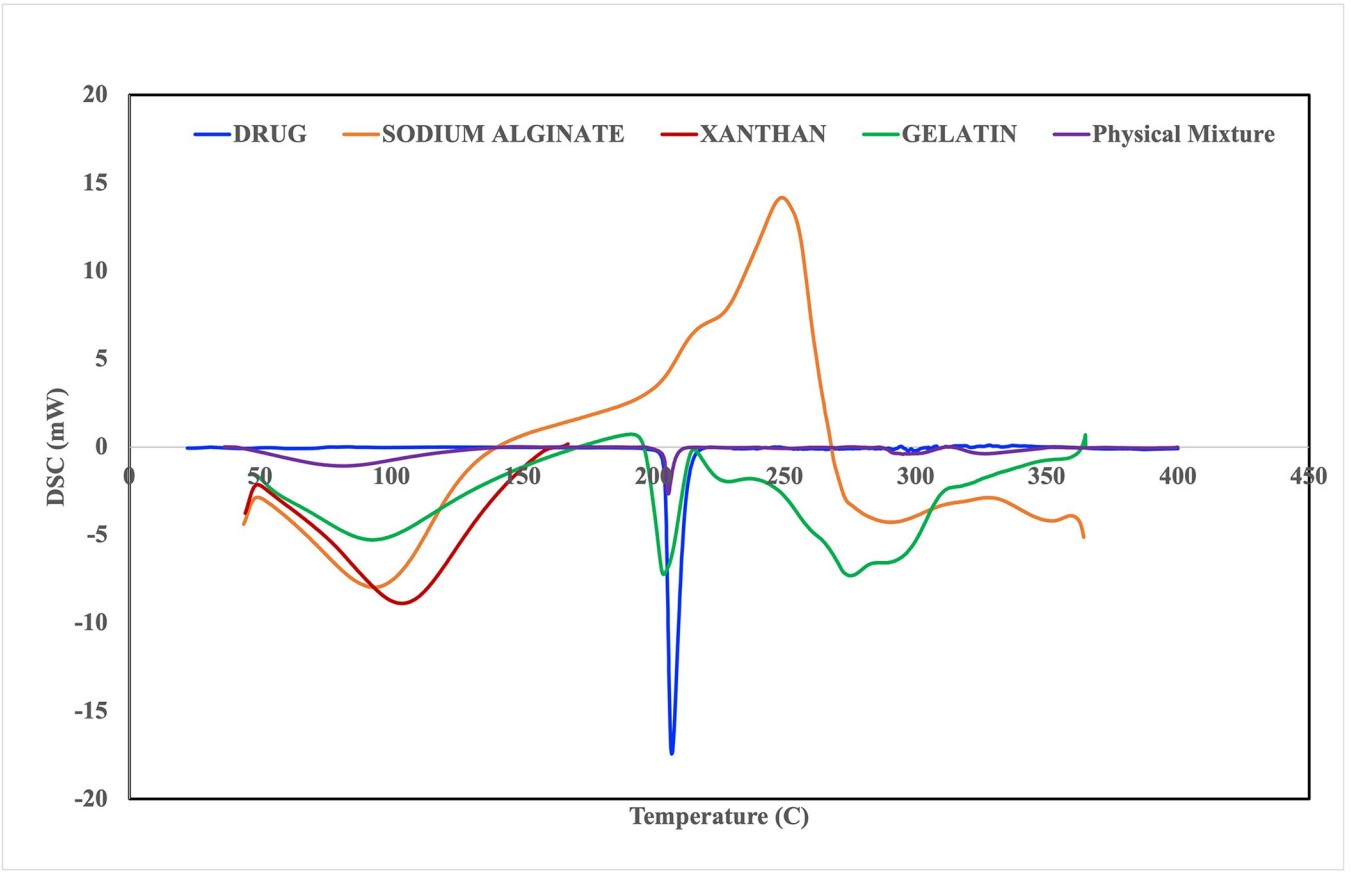

**Fig 1. DSC thermograms of drug and excipients.**

prepared from xanthan gum, sodium alginate, and pullulan as polymers, and 10% w/w propylene glycol as a plasticizer that gave the best film flexibility. According to the results in Table 2.

**3.2.2. Film thickness.** The thickness of the prepared films is illustrated in Table 2. The results demonstrate that all Eszopiclone FDF formulas showed a thickness from 0.23–0.28 mm. This uniform thickness assures the uniform distribution of the drug and polymer inside the film, causing the uniformity in weight and drug content (uniform dosing). The variation in thickness is less than ± 5%.

**Table 2. *In vitro* evaluation of medicated fast dissolving films (FDF).**

| Formula N° | Visual examination | Thickness (mm) | Surface pH range | Drug content | Tensile strength MPa | Elongation% EB % | Folding endurance | Disintegration time (sec) |
|---|---|---|---|---|---|---|---|---|
| F1 | ++ | 0.23 ± 0.01 | 6.7 | 108 ± 0.03 | 16.18 ± 0.02 | 27.38 ± 0.01 | 281 ± 5 | 11 ± 0.02 |
| F2 | +++ | 0.24 ± 0.01 | 6.7 | 91 ± 0.01 | 20.72 ± 0.02 | 36.43 ± 0.02 | 296 ±2 | 13 ± 0.02 |
| F3 | ++ | 0.24 ± 0.02 | 6.5 | 99 ± 0.04 | 19.87 ± 0.02 | 26.88 ± 0.03 | 269 ± 2 | 12 ± 0.01 |
| F4 | +++ | 0.25 ± 0.01 | 6.8 | 97 ± 0.03 | 25.74 ± 0.02 | 35.23 ± 0.03 | 291 ±1 | 14 ± 0.03 |
| F5 | + | 0.26 ± 0.04 | 6.6 | 109 ± 0.03 | 11.12 ± 0.01 | 25.38 ± 0.05 | 200 ±2 | 9 ± 0.02 |
| F6 | +++ | 0.28 ± 0.02 | 6.6 | 103 ± 0.03 | 15.67 ± 0.01 | 30.41 ± 0.04 | 253 ±1 | 10 ± 0.01 |
| F7 | ++ | 0.27 ± 0.01 | 6.8 | 99 ± 0.01 | 12.99 ± 0.01 | 28.11 ± 0.04 | 290 ±3 | 35 ± 0.01 |
| F8 | ++ | 0.23 ± 0.05 | 6.5 | 100 ± 0.02 | 14.41 ± 0.02 | 32.89 ± 0.01 | 300 ±4 | 40 ± 0.05 |

**3.2.3. The pH of the surface.** The surface **pH** of the prepared films ranged from 6.5–6.8 within the physiological range, as indicated in Table 2. Therefore, the prepared films are nonirritant to the lung [33].

**3.2.4. Drug content.** According to the results indicated in Table 2. All Eszopiclone FDFs formulas showed a drug content range from 97%–109% as F5 > F1 > F6 > F8 > F3 = F4 > F7 > F2. Each 1cm$^2$ film contained ~ 1-mg Eszopiclone as expected.

## 3.3. Mechanical Invitro evaluation of fast dissolving films (FDF)

**3.3.1. Tensile strength (TS), percentage elongation (EB%).** The ideal fast dissolving film has higher tensile strength and percentage elongation. [34]. Table 2 indicates the TS values and EB%, which differed from 11.12 to 25.74 (MPa) and 25.38%–36.43%, respectively. By increasing the plasticizer concentration from (5% to 10%), TS and EB% increase. The addition of pullulan decreases the compatibility of film components, and therefore, film flexibility reduced decreasing in EB% and FE [35]. Pullulan loses its water content easily during the drying process, so it causes less bonding, more phase separation, and larger particle sizes reducing EB%. An increase in particle size causes a loss of mechanical qualities [27]. But xanthan gum, gelatin, and sodium alginate tend to be gelatinous and retain water. According to the literature, TS should be maximized, EB% should be about 20% [36].

**3.3.2. Folding endurance.** A higher FE value indicates the more mechanical strength of a film. A direct relation exists between mechanical strength and FE of films. The FE varied between 200 and 300 times, as indicated in Table 2, which is considered the sign of good flexibility and elasticity. Mechanical strength is governed by a plasticizer concentration, which directly affects FE value [24]. Also, the FE was increased as the concentration of the polymer increased.

## 3.4. In vitro disintegration time (DT) test of fast dissolving films (FDF)

*In vitro* disintegration time (DT) was determined in a phosphate buffer (**pH** 6.8) as shown in Table 2. The results clearly indicated that the formulated films had a fast disintegrating nature at saliva **pH**. The polymers used to prepare films had good film-forming properties with good wetting properties [37]. F4 (3% sodium alginate and 10% propylene glycol) show the fast DT while F8 (2.5% gelatin and 10% propylene glycol) show the delayed DT 40 sec therefore, it was excluded from *in vivo* study. Also, Plasticizer act by inserting themselves between the polymer strands and breaking the polymer-polymer interactions, and increasing the molecular mobility of the polymer strands [38], which plays a role in increasing the disintegration. It was concluded that polymer affects the disintegration time.

## 3.5. In vitro dissolution test of fast dissolving films (FDF)

The release study was conducted using a phosphate buffer (**pH** 6.8) to simulate the condition of the buccal cavity. Dissolution samples were taken at 1, 2, 3, and 7 min. The drug release was in the range of 78.51%–99.99%. All polymers used are hydrophilic polymers that absorb the water and form the gel responsible for the retardation of the drug from the film system. Hydrophilic polymers create the pore and channels on the film surface that released the drug from the film [39]. The results of the dissolution test are illustrated in Fig 2.

## 3.6. Taste masking test (human panel testing) of fast dissolving films (FDF)

The drug was very bitter; all three trilaminated formulations are pleasant. F6 > F4 > F2 > Tablet. These results due to the polysaccharide properties of pullulan [40]. The drug does

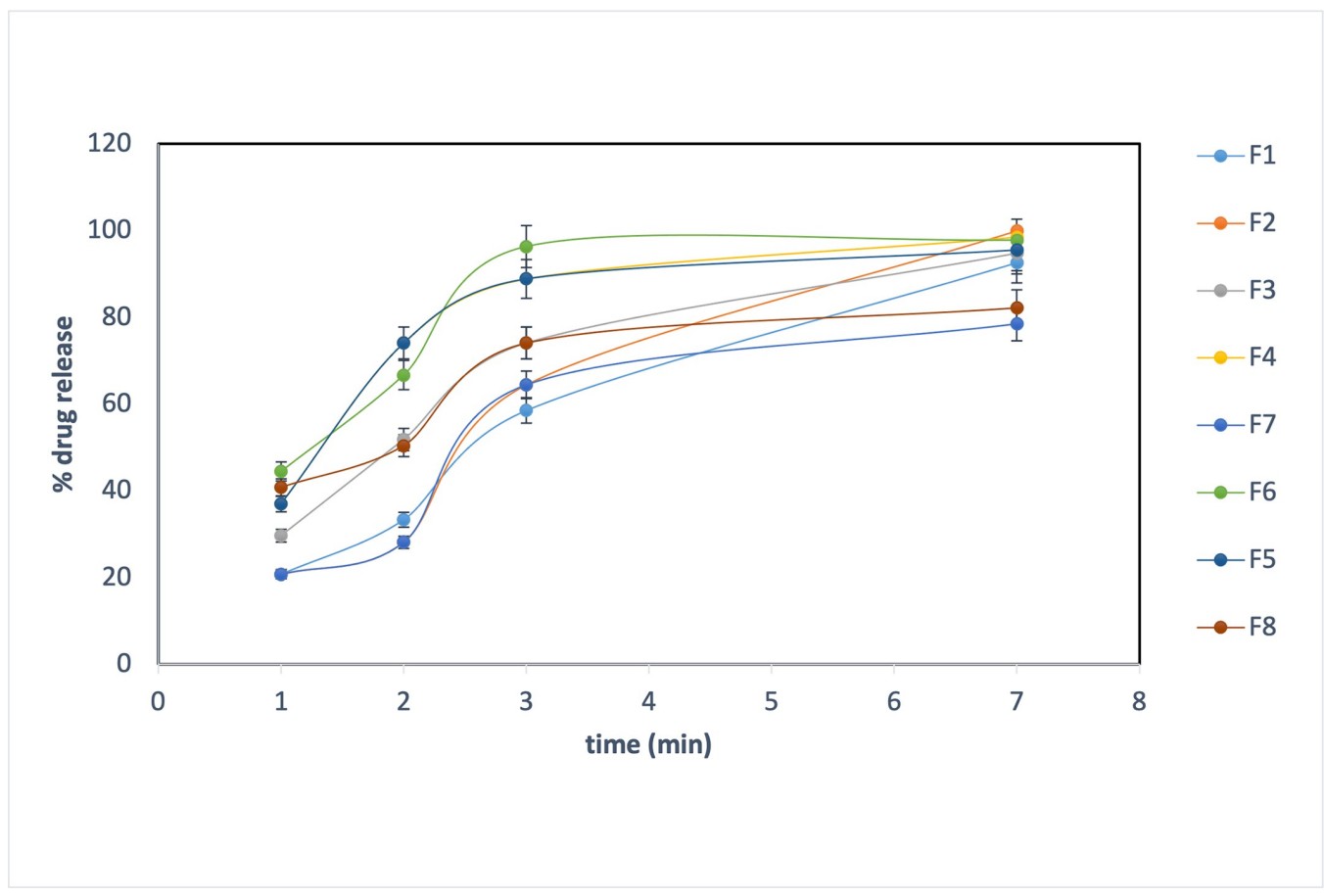

**Fig 2. *In vitro* dissolution in phosphate buffer pH 6.8.** F7 (2.5% gelatin and 5% propylene glycol) have the lowest percentage drug release within the first seven minutes (78.51%). The percentage of drug released within the first seven minutes from F2, F6, and F2 and formula containing 10% propylene glycol shows higher drug release than other formula (97.7%, 98.52% 99.99%, respectively).

not move from its layer unless it has been wetted with water. So, special packaging is required for products stability and safety [41]. The saliva in the oral cavity wets the film as it consists of hydrophilic polymers and dissolves it rapidly, drug at this point migrate from its layer to sweetening layer so the bitter taste of the drug was masked completely by this technique.

## 3.7. In vivo pharmacokinetic studies of fast dissolving films (FDF) in human volunteers

The optimized formulations (F2, F4, and F6) were used for *in vivo* evaluation compared to a marketed Eszopiclone 1-mg tablet (Lunesta ®). The plasma concentration of Eszopiclone versus time for all samples was plotted in Fig 3C.

A one-way analysis of variance (ANOVA) followed by the least significant difference (LSD) as a post hoc test was applied; using SPSS program version 17 software. The differences were considered significant if $P < 0.05$. The column effluent was detected spectrophotometrically at 304 nm. Retention time for aripiprazole was 1.929 min which is well differentiated from the peak of the eszopiclone drug which has retention time 1.865 min as shown in Fig 3B which also revealed that there were no peaks due to formula compounds that might interfere with the assay. Calibration curve were constructed for eszopiclone with an equation that best describe

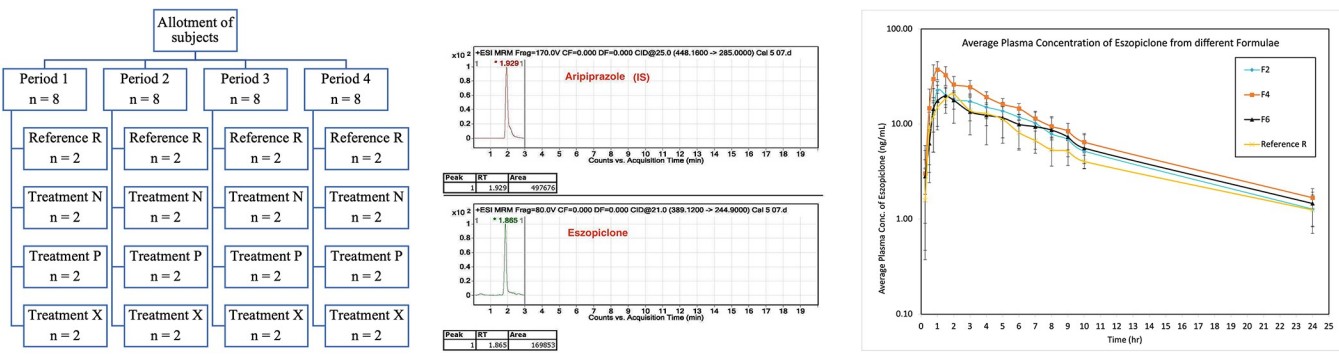

**Fig 3. a.** Design of Pharmacokinetics study. **b.** LC-MS_MS chromatograms of Eszopiclone. **c.** Average plasma drug level of Eszopiclone.

the curve is (y = 0.0137 x– 0.0102) as y is represented by peak area and x for the concentration in (ng/ml) ($r^2$ = 0.9774) as shown in Fig 3D. The limit of detection and the limit of quantitation were 0.05 and 0.5 (ng/ml) respectively.

Key pharmacokinetic parameters, peak plasma concentration ($C_{max}$), time to reach the maximum peak ($T_{max}$), and the extent of absorption (AUC) for the three Eszopiclone FDF and lunesta® were calculated in Table 3. A significant difference (p-value < 0.05) was found between the reference and F4 which contain sodium alginate as polymer in all parameters tested as seen in the sheet attached with our response. Tmax is decreased in F4 formulae which contain sodium alginate as polymer and it was 1.063 h for F4 while 2.25 h only for commercial tablet. Relative bioavailability is increased 1.6 fold for F4, 1.2 fold for F2 and 1.1 fold for F6.

Eszopiclone FDF containing sodium alginate as polymer indicated the highest $C_{max}$ (39.741 ± 6.785-μg/l) and the lowest $T_{max}$ (1.063 h) than Eszopiclone FDF containing xanthan, pullulan as polymers, which showed $C_{max}$ (24.862 ± 2.805-μg/l and 24.89 ± 7.838-μg/l) respectively and $T_{max}$ (1.188 h and 1.344) and the Eszopiclone ® (with $C_{max}$ = 22.132 ± 2.107-μg/l and $T_{max}$ = 2.25 h), showing a significant difference in their absorption rate. This could be attributed to the faster disintegration rate of FDFs among the conventional sublingual tablets. The calculated AUC values for Eszopiclone FDFs containing sodium alginate, xanthan, pullulan as polymers and Eszopiclone ® were 241.788 ± 19.053-μgh/l, and 180.572 ± 17.31-μg h/l, 175.225 ± 61.746-μg h/l, and 151.92 ± 28.02, respectively, as indicated in Table 3.

The increased AUC showed a better absorption extent of sodium alginate containing FDF than xanthan FDF, pullulan FDF, and conventional tablets. The rate and extent of drug absorption were enhanced by the FDF technique, considering the elimination of the influences of the gastrointestinal tract and the first-pass effect through the buccal trans-mucosal absorption [42, 43].

## 4. Conclusion

In this study, Eszopiclone was incorporated for the first time in trilaminated FDF using various polymers: pullulan sodium alginate, xanthan gum, and gelatin. Based on the obtained results,

**Table 3. *In vivo* pharmacokinetic parameters & data represent the mean value ± standard deviation (SD).**

| Treatment | Cmax (ng/mL) | Tmax (hr) | AUC0_t (hr*ng/mL) | AUMC0_t (hr*hr*ng/mL) | AUMC0_INF (hr*hr*ng/mL) |
|---|---|---|---|---|---|
| REFRENCE | 22.132 ± 2.107 | 2.25 ± 1.134 | 138.071 ± 22.571 | 900.696 ± 207.04 | 1389.48 ± 469.678 |
| F2 | 24.862 ± 2.805 | 1.188 ± 0.259 | 168.669 ± 20.204 | 1104 ± 124.554 | 1507.125 ± 188.939 |
| F4 | 39.741 ± 6.785 | 1.063 ± 0.291 | 225.69 ± 19.68 | 1402.216 ± 139.466 | 1948.783 ± 204.245 |
| F6 | 24.89 ± 7.838 | 1.344 ± 0.399 | 160.345 ± 58.375 | 1123.805 ± 408.136 | 1641.089 ± 620.144 |

trilaminated FDF masks Eszopiclone's bitter taste with acceptable uniformity content, film thickness, percentage dissolution, disintegration time, FE, EB%, and TS properties. Formulations with F2 (0.05% xanthan gum, 10% propylene glycol), F4 (3% sodium alginate, 10% propylene glycol), and F6 (2.5% pullulan, 10% propylene glycol) were represented as optimized formulations for *in vivo* study. Pharmacokinetic studies in humans demonstrated that F4 containing sodium alginate as polymer had the highest AUC and $C_{max}$ as well as a lower $T_{max}$ among other formulations and conventional tablets. Therefore, FDFs' technology could improve the therapeutic effect of Eszopiclone.

## Author Contributions

**Conceptualization:** Mahmoud Teaima, Mohamed Yasser, Kamel Shoueir, Mohamed El-Nabarawi, Doaa Helal.

**Methodology:** Nehal Elfar.

**Supervision:** Mahmoud Teaima, Mohamed Yasser, Kamel Shoueir, Mohamed El-Nabarawi, Doaa Helal.

**Writing – original draft:** Nehal Elfar.

**Writing – review & editing:** Mahmoud Teaima, Mohamed Yasser, Kamel Shoueir, Mohamed El-Nabarawi, Doaa Helal.

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
