## [Decision Letter · Decision Letter 0]

28 Jan 2022

PONE-D-21-37028Construction of Sublingual Trilaminated Eszopiclone Fast Dissolving Film for Treatment of Insomnia: Formulation, Characterization and In-vivo Clinical Comparative Pharmacokinetic Study in Healthy Human Subjects.PLOS ONE

Dear Dr. Teaima,

Thank you for submitting your manuscript to PLOS ONE. After careful consideration, we feel that it has merit but does not fully meet PLOS ONE’s publication criteria as it currently stands. Therefore, we invite you to submit a revised version of the manuscript that addresses the points raised during the review process.

We look forward to receiving your revised manuscript.

Kind regards,

José das Neves

Academic Editor

PLOS ONE

Journal Requirements:

2. Please amend your Methods section to state the name of your Institutional Review Board and to state any inclusion/exclusion criteria for participants.

- https://pubmed.ncbi.nlm.nih.gov/24015378/

In your revision ensure you cite all your sources (including your own works), and quote or rephrase any duplicated text outside the methods section. Further consideration is dependent on these concerns being addressed.

4. We note that you have stated that you will provide repository information for your data at acceptance. Should your manuscript be accepted for publication, we will hold it until you provide the relevant accession numbers or DOIs necessary to access your data. If you wish to make changes to your Data Availability statement, please describe these changes in your cover letter and we will update your Data Availability statement to reflect the information you provide

Reviewers' comments:

Reviewer's Responses to Questions

**Comments to the Author**

1. Is the manuscript technically sound, and do the data support the conclusions?

Reviewer #1: Partly

Reviewer #2: Partly

2. Has the statistical analysis been performed appropriately and rigorously? 

Reviewer #1: No

Reviewer #2: No

3. Have the authors made all data underlying the findings in their manuscript fully available?

Reviewer #1: No

Reviewer #2: No

4. Is the manuscript presented in an intelligible fashion and written in standard English?

Reviewer #1: No

Reviewer #2: Yes

5. Review Comments to the Author

Reviewer #1: Few comments are as follows:

1) pH word in abstract should be corrected.

2) Heading for 2.2.2, what is tril?

3) Please explain rational for selections of polymer concentration. It is based on Literature search or placebo trials.

4) 2.2.4 : In which layer and when drug should be added is not mentioned.

5) 2.2.5, sandwich approach is good but is drug migration occurs from middle to end layer?

6) P from petri plates should be small at many places.

7) Is for testing in volunteers any ethical approval taken?

8) Why ariprirazole is selected as IS?

9) Line 276, page 14. Gives characteristics of buccal film. Is film described in this manuscript buccal or dissolving?

10) Line 317, page 16, says results are due to polysaccharide properties of pullulan. But maximum good taste is shown by F4 having sodium alginate as polymer. Explain.

11) Figure 1 showed DSC thermogram of excipients and 1 physical mixture. It is physical mixture with which excipient? For better understanding of interaction, authors should show physical mixtures of drug with all other polymers.

12) LC-MS/MS chromatogram of drug and IS should be shown.

13) Statistical analysis is explained but not seen in discussion , in figures please include.

Reviewer #2: This is an interesting paper about formulation of eszopiclone oral film. The paper contains manufacturing of films, in vitro evaluations and test in healthy volunteers. The paper is designed suitably. The main concern is that films disintegrated over 10 minutes, much longer than fast dissolving oral films. These cannot be called fast dissolving oral films, as this type of films disintegrate less than one minute. Therefore, with this long disintegration time, how can they be beneficial to patients? The comments are as the following:

1. There are paragraphs in the Introduction that do not cite references such as the last paragraph in page 3. Also, there is an incomplete sentence on page 3 line 60, and line 68, “better” should be replaced with “bitter”.

2. Could the authors please specify the calibration curve characteristics of eszopiclone for UV measurements on page 8.

3. Were ethics obtained to test the taste of films in volunteers and what were exclusion and inclusion criteria?

4. It is claimed on page 11, Line 225 that the LC-MS data were validated, where is the data or reference? What were the limit of detection and limit of quantification?

5. Please use past tense for line 257.

6. Line 284, please cite a reference.

7. How did the volunteers hold a film for 10 minutes in the mouth, did they chew them and then swallow?

8. Did the authors obtain GMP grades of active ingredient and excipients to prepare the oral films for volunteers? Who supplied these?

9. A statistical method is described in the Method section but the results are not presented.

6. PLOS authors have the option to publish the peer review history of their article (what does this mean?). If published, this will include your full peer review and any attached files.

Reviewer #1: No

Reviewer #2: No

---

## [Author Response · Author response to Decision Letter 0]

20 Feb 2022

Peer Reviewer: 1

Comments to the Author: 

1- pH word in abstract should be corrected.

Response: The mistake has been corrected throughout the manuscript and highlighted with yellow color

2- Heading for 2.2.2, what is tril?

Response: it is a written mistake and the correct word is (triple). The correct word has been written in the revised manuscript.

3- Please explain rational for selections of polymer concentration. It is based on Literature search or placebo trials.

Response:

We tried many different concentrations of many polymers in the preliminary study. From which the best results either type of polymer or polymer concentration were selected for construction of eszopiclone triple fast dissolving film. Different researchers in the literature used our polymers in different concentration [1-6].in our work we used this concentrations and additional concentrations in the preliminary study.

Therefore the rational is based on both literature and placebo trials.

The tables of the preliminary study:

formula polymer type polymer conc Plasticizer type Plasticizer conc solvent type

F1 gelatin 40% Propylene Glycol 10% Methanol:Methylene Chloride (1:1)

F2 gelatin 40% Glycerin 20% Isopropyl alcohol:H2O (1:4)

F3 casein 1.50% Glycerin 10% Methanol:Methylene Chloride (1:1)

F4 casein 1.50% Glycerin 10% Isopropyl alcohol:H2O (1:4)

F5 HPMC 1% Propylene Glycol 10% Methanol:Methylene Chloride (1:1)

F6 HPMC 1% Glycerin 20% Methanol:Methylene Chloride (1:1)

F7 gelatin 40% Glycerin 10% Isopropyl alcohol:H2O (1:4)

F8 HPMC 1% Propylene Glycol 20% Isopropyl alcohol:H2O (1:4)

F9 gelatin 40% Propylene Glycol 30% Isopropyl alcohol:H2O (1:4)

F10 gelatin 40% Propylene Glycol 20% Methanol:Methylene Chloride (1:1)

F11 casein 1.50% Glycerin 30% Methanol:Methylene Chloride (1:1)

F12 HPMC 1% Propylene Glycol 30% Methanol:Methylene Chloride (1:1)

F13 gelatin 40% Glycerin 30% Methanol:Methylene Chloride (1:1)

F14 HPMC 1% Glycerin 10% Isopropyl alcohol:H2O (1:4)

F15 casein 1.50% Propylene Glycol 30% Isopropyl alcohol:H2O (1:4)

F16 casein 1.50% Propylene Glycol 10% Isopropyl alcohol:H2O (1:4)

F17 HPMC 1% Glycerin 30% Isopropyl alcohol:H2O (1:4)

F18 casein 1.50% Propylene Glycol 20% Methanol:Methylene Chloride (1:1)

F19 casein 1.50% Propylene Glycol 10% Water

F20 HPMC 2% Propylene Glycol 15% Methanol:Methylene Chloride (1:1)

formula polymer type polymer conc Plasticizer type Plasticizer conc solvent type

F21 casein 1.50% Glycerin 10% Methanol:Methylene Chloride (1:1)

F22 PVA 10% Glycerin 5% Water

F23 casein 1.50% Propylene Glycol 5% Methanol:Methylene Chloride (1:1)

F24 PVA 10% Glycerin 5% Methanol:Methylene Chloride (1:1)

F25 casein 1.50% Glycerin 5% Isopropyl alcohol:H2O (1:4)

F26 PVA 10% Glycerin 15% Methanol:Methylene Chloride (1:1)

F27 casein 1.50% Propylene Glycol 10% Isopropyl alcohol:H2O (1:4)

F28 HPMC 2% Glycerin 10% Water

F29 HPMC 2% Glycerin 15% Isopropyl alcohol:H2O (1:4)

F30 HPMC 2% Propylene Glycol 5% Water

F31 PVA 10% Propylene Glycol 5% Isopropyl alcohol:H2O (1:4)

F32 PVA 10% Glycerin 15% Isopropyl alcohol:H2O (1:4)

F33 casein 1.50% Glycerin 15% Water

F34 HPMC 2% Glycerin 5% Methanol:Methylene Chloride (1:1)

F35 HPMC 2% Propylene Glycol 15% Water

F36 casein 1.50% Propylene Glycol 15% Methanol:Methylene Chloride (1:1)

F37 HPMC 2% Propylene Glycol 10% Isopropyl alcohol:H2O (1:4)

F38 HPMC 2% Propylene Glycol 10% Methanol:Methylene Chloride (1:1)

F39 PVA 10% Propylene Glycol 10% Water

F40 PVA 10% Glycerin 10% Isopropyl alcohol:H2O (1:4)

formula polymer type polymer conc Plasticizer type Plasticizer conc solvent type

F41 HPMC 2% Propylene Glycol 10% ,Isopropyl alcohol:H2O (1:4)

F42 HPMC 2% Glycerin 5% Isopropyl alcohol:H2O (1:4)

F43 PVA 10% Propylene Glycol 10% Water

F44 PVA 10% Glycerin 5% Water

F45 HPMC 2% Propylene Glycol 5% Isopropyl alcohol:H2O (1:4)

F46 HPMC 2% Glycerin 5% Water

F47 PVA 10% Glycerin 5% Isopropyl alcohol:H2O (1:4)

F48 HPMC 2% Glycerin 10% Isopropyl alcohol:H2O (1:4)

F49 HPMC 2% Propylene Glycol 5% Water

F50 HPMC 2% Glycerin 10% Water

F51 PVA 10% Propylene Glycol 5% Water

F52 PVA 10% Propylene Glycol 5% Isopropyl alcohol:H2O (1:4)

F53 HPMC 2% Propylene Glycol 10% Water

F54 PVA 10% Propylene Glycol 10% Isopropyl alcohol:H2O (1:4)

F55 PVA 10% Glycerin 10% Water

F56 PVA 10% Glycerin 10% Isopropyl alcohol:H2O (1:4)

F57 HPMC:Na alginate (1:1) propylene glycol 5% Isopropyl alcohol:H2O (1:4)

F58 PVA 2% propylene glycol 15% Isopropyl alcohol:H2O (1:4)

F59 Pollulan 2% Glycerin 5% Isopropyl alcohol:H2O (1:4)

F60 Pollulan:Maltodextrin (1:1) Glycerin 5% Isopropyl alcohol:H2O (1:4)

F61 HPMC 2% propylene glycol 5% Isopropyl alcohol:H2O (1:4)

F62 HPMC:Pollulan (1:1) Glycerin 5% Isopropyl alcohol:H2O (1:4)

F63 HPMC:Pollulan (1:1) Glycerin 15% Isopropyl alcohol:H2O (1:4)

F64 PVA 2% propylene glycol 15% Isopropyl alcohol:H2O (1:4)

F65 PVA 2% propylene glycol 10% Isopropyl alcohol:H2O (1:4)

F66 HPMC:Na alginate (1:1) Glycerin 15% Isopropyl alcohol:H2O (1:4)

F67 PVA 2% Glycerin 10% Isopropyl alcohol:H2O (1:4)

F68 PVA propylene glycol 5% Isopropyl alcohol:H2O (1:4)

F69 HPMC:Pollulan (1:1) propylene glycol 15% Isopropyl alcohol:H2O (1:4)

F70 HPMC:PVA (1:1) Glycerin 5% Isopropyl alcohol:H2O (1:4)

F71 HPMC:Na alginate (1:1) propylene glycol 15% Isopropyl alcohol:H2O (1:4)

F72 HPMC:PVA (1:1) propylene glycol 5% Isopropyl alcohol:H2O (1:4)

F73 HPMC:PVA (1:1) propylene glycol 15% Isopropyl alcohol:H2O (1:4)

F74 HPMC 2% Glycerin 5% Isopropyl alcohol:H2O (1:4)

F75 HPMC:PVA (1:1) propylene glycol 5% Isopropyl alcohol:H2O (1:4)

F76 PVA 2% Glycerin 5% Isopropyl alcohol:H2O (1:4)

F77 HPMC 2% propylene glycol 15% Isopropyl alcohol:H2O (1:4)

F78 HPMC:PVA (1:1) propylene glycol 10% Isopropyl alcohol:H2O (1:4)

F79 HPMC 2% Glycerin 15% Isopropyl alcohol:H2O (1:4)

F80 pullulan 2% propylene glycol 5% Isopropyl alcohol:H2O (1:4)

formula polymer type polymer conc Plasticizer type Plasticizer

 conc solvent type

F81 Pollulan:Maltodextrin (1:1) propylene glycol 15% Isopropyl alcohol:H2O (1:4)

F82 HPMC:Pollulan (1:1) propylene glycol 5% Isopropyl alcohol:H2O (1:4)

F83 HPMC:PVA (1:1) Glycerin 15% Isopropyl alcohol:H2O (1:4)

F84 Pollulan:Maltodextrin (1:1) Glycerin 15% Isopropyl alcohol:H2O (1:4)

F85 pullulan 2% Glycerin 15% Isopropyl alcohol:H2O (1:4)

F86 HPMC:PVA (1:1) Glycerin 10% Isopropyl alcohol:H2O (1:4)

F87 Pollulan:Maltodextrin (1:1) propylene glycol 5% Isopropyl alcohol:H2O (1:4)

F88 pullulan 2% propylene glycol 15% Isopropyl alcohol:H2O (1:4)

F89 PVA 2% Glycerin 15% Isopropyl alcohol:H2O (1:4)

F90 HPMC:Na alginate (1:1) Glycerin 5% Isopropyl alcohol:H2O (1:4)

F91 HPMC 2% propylene glycol 5% Isopropyl alcohol:H2O (1:4)

F92 HPMC 2% propylene glycol 15% Isopropyl alcohol:H2O (1:4)

F93 HPMC 2% propylene glycol 10% Isopropyl alcohol:H2O (1:4)

F94 HPMC , Sodium alginate ,Pullulan 1%,1%,1% propylene glycol 15% Isopropyl alcohol:H2O (1:4)

F95 HPMC , Sodium alginate ,Pullulan 1%,1%,1% propylene glycol 10% Isopropyl alcohol:H2O (1:4)

F96 HPMC , Sodium alginate ,Pullulan 1%,1%,1% propylene glycol 5% Isopropyl alcohol:H2O (1:4)

F97 Sodium alginate 2% propylene glycol 15% Isopropyl alcohol:H2O (1:4)

F98 Sodium alginate 2% propylene glycol 5% Isopropyl alcohol:H2O (1:4)

F99 Sodium alginate 2% propylene glycol 10% Isopropyl alcohol:H2O (1:4)

F100 Pullulan 2% propylene glycol 10% Isopropyl alcohol:H2O (1:4)

formula polymer type polymer conc Plasticizer type Plasticizer

 conc solvent type

F101 Pullulan 2% propylene glycol 15% Isopropyl alcohol:H2O (1:4)

F102 Pullulan 2% propylene glycol 5% Isopropyl alcohol:H2O (1:4)

F103 Pullulan , HPMC 1.5%, 0.5 % propylene glycol 10% Isopropyl alcohol:H2O (1:4)

F104 Pullulan , HPMC 1%, 1% propylene glycol 10% Isopropyl alcohol:H2O (1:4)

F105 Pullulan , HPMC 1%, 2% propylene glycol 10% Isopropyl alcohol:H2O (1:4)

F106 Pullulan , Maltodextrin 2%,1% propylene glycol 10% Isopropyl alcohol:H2O (1:4)

F107 Pullulan , HPMC 1.5%, 1 % propylene glycol 10% Isopropyl alcohol:H2O (1:4)

F108 Pullulan , Maltodextrin 1.5%,1% propylene glycol 10% Isopropyl alcohol:H2O (1:4)

F109 

F110 Pullulan , Maltodextrin 2.5%,2% propylene glycol 10% Isopropyl alcohol:H2O (1:4)

F111 Pullulan 5% propylene glycol 10% Isopropyl alcohol:H2O (1:4)

F112 Pullulan , HPMC 2.5%,2.5% propylene glycol 10% Isopropyl alcohol:H2O (1:4)

F113 HPMC, Xanthan gum 2.5%, 1% propylene glycol 10% Isopropyl alcohol:H2O (1:4)

F114 Pullulan, Xanthan gum 2% ,2% propylene glycol 10% Isopropyl alcohol:H2O (1:4)

F115 Pullulan 1.50% propylene glycol 10% Isopropyl alcohol:H2O (1:4)

F116 Xanthan gum 0.5% propylene glycol 10% Isopropyl alcohol:H2O (1:4)

F117 Sodium alginate 3% propylene glycol 10% Isopropyl alcohol:H2O (1:4)

4- 2.2.4 : In which layer and when drug should be added is not mentioned.

Response: first we prepare the three films separately:

1) The first film consists of flavoring agent was prepared using casting method in which flavoring agent (clove oil and peppermint oil), HPMC as film forming polymer, propylene glycol as plasticizer, casting solvent (distilled water:isopropyl alcohol (1:1)) were used.

2) The second film consists of the drug was prepared using casting method in which drug (eszopiclone) ,( xanthan gum,sodium alginate or pullulan ) as film forming polymer, propylene glycol as plasticizer, casting solvent (distilled water:isopropyl alcohol (1:1)) were used.

3) The third film consists of sweetening agent was prepared using casting method in which sweetening agent (sucralose), HPMC as film forming polymer, propylene glycol as plasticizer, casting solvent (distilled water:isopropyl alcohol (1:1)) were used.

4) After drying of the three films we stick the first film (flavoring film) and the third film (sweetening film) in the both sides of the second film (drug containing film) using small drops of casting solvent. As mentioned in 2.2.5 and in the figure below.

 

5- 2.2.5, sandwich approach is good but is drug migration occurs from middle to end layer?

Response:

The saliva in the oral cavity wets the film as it consists of 

hydrophilic polymers and dissolves it rapidly. Thus the drug is 

released into the saliva and is absorbed via the highly 

vascularized oro mucosal tissues.

The drug does not move from its layer unless it has been wetted with water. So it is Require special packaging for products stability and safety[7].The saliva in the oral cavity wets the film as it consists of hydrophilic polymers and dissolves it rapidly. Thus the drug is released into the saliva and is absorbed via the highly vascularized oro mucosal tissues[7]. The discussion was enriched by your valuable comments in section 3.6 page 16 in the revised manuscript.

6- P from petri plates should be small at many places.

Response: All words have been revised, corrected and highlighted.

7- Is for testing in volunteers any ethical approval taken?

The affirmed protocol set by the ethics committee of the Faculty of Pharmacy, Cairo University, Egypt with serial number: PI (2313) and the date of this approval (26 / 11 /2018).

8- Why ariprirazole is selected as IS?

Response: ariprirazole was selected as IS to enhance the calculation using the peak area ratio instead of peak area only to cancel the error due to injection or the error sampling on HPLC and the method was attached in the mail. 

Ariprirazole according to its chemical structure has close polarity to our drug (eszopiclone). So it is peak appeared at a retention time 1.929 which is well differentiated from the peak of the drug which has retention time 1.865 as seen in the figure 3b.

 Figure 3b: sample of Chromatogram of Eszopiclone (drug) and Aripiprazolr (IS)

Figure of aripiprazole chemical structure

Figure of eszopiclone chemical structure

9- Line 276, page 14. Gives characteristics of buccal film. Is film described in this manuscript buccal or dissolving?

Response:

Formula was prepared as fast dissolving film to achieve higher plasma concentration at lower tmax and also to decrease the residence time in mouth to enhance patient compliance for this drug regarding its bitter taste so buccal film was written as representation of the mouth area. So to correct this misunderstanding, the word buccal has been replaced with fast dissolving film in line 276 page 14.

10- Line 317, page 16, says results are due to polysaccharide properties of pullulan. But maximum good taste is shown by F4 having sodium alginate as polymer. Explain.

Response:

The preliminary study consists of 117 formula from which we choose the best eight formula to continue the advanced tests. In line 317 page 16 the point 3.6 taste masking test the pullulan was given better taste than sodium alginate. Because of this result, we searched for a reason and found that the pullulan has polysaccharide properties so this arrangement was overlapped by mistake and the correct arrangement is F6 > F4> F2 > tablet. This mistake was corrected in page 16, line 316 and highlighted.

11- Figure 1 showed DSC thermogram of excipients and 1 physical mixture. It is physical mixture with which excipient? For better understanding of interaction, authors should show physical mixtures of drug with all other polymers.

Response:

Figure 1: DSC 

The physical mixture containing (eszopiclone: pullulan: sodium alginate: xanthan gum: gelatin) in a ratio (1:1:1:1:1) w/w.

12- LC-MS/MS chromatogram of drug and IS should be shown.

Response: 

The chromatogram was attached in the main manuscript line 329 page 17.

Fig 3b. LC-MS_MS chromatograms of Eszopiclone.

13- Statistical analysis is explained but not seen in discussion, in figures please include.

Response:

All statistical results are attached with the revised manuscript. 

A one-way analysis of variance (ANOVA) followed by the least significant difference (LSD) as a post hoc test was applied; using SPSS program version 17 software. The differences were considered significant if P<0.05. The column effluent was detected spectrophotometrically at 304 nm. Retention time for aripiprazole was 1.929 min which is well differentiated from the peak of the eszopiclone drug which has retention time 1.865 min as shown in figure (3b) which also revealed that there were no peaks due to formula compounds that might interfere with the assay. Calibration curve were constructed for eszopiclone with an equation that best describe the curve is (y = 0.0137 x – 0.0102 ) as y is represented by peak area and x for the concentration in (ng/ml) (r2 = 0.9774 ) as shown in figure (3d).The limit of detection and the limit of quantitation were 0.05 and 0.5 (ng/ml) respectively.

Discussion: When we compare the reference tablet with the three fast dissolving films containing sodium alginate or xanthan gum or pullulan as polymer to study its effect on Cmax , AUC 0-24 and AUC 0-INF. A significant difference (p-value < 0.05) was found between the reference and F4 which contain sodium alginate as polymer in all parameters tested as seen in the sheet attached with our response. Tmax is decreased in F4 formulae which contain sodium alginate as polymer and it was 1.063 h for F4 while 2.25 h only for commercial tablet . Relative bioavailability is increased 1.6 fold for F4, 1.2 fold for F2 and 1.1 fold for F6.

Peer Reviewer: 2

• The main concern is that films disintegrated over 10 minutes, much longer than fast dissolving oral films. 

Response:

 It is typing error and the correct time is (sec). The correct time has been written in the revised manuscript. 

The disintegration time of all formula were found to be between 10 to 40 sec so it is called fast dissolving film. And seen in Table 2 (In vitro evaluation of medicated fast dissolving films (FDF)), page 14 in the main manuscript.

1. There are paragraphs in the Introduction that do not cite references such as the last paragraph in page 3. Also, there is an incomplete sentence on page 3 line 60, and line 68, “better” should be replaced with “bitter”.

Response:

a. We revised the manuscript and the references were added to uncited paragraph and highlighted with yellow color. Additional references were added to other paragraphs which lack of references such as line 154,156,159.

b. The incomplete sentence was revised and completed.

c. “Better” word has been replaced with “bitter” in line 69.

2. Could the authors please specify the calibration curve characteristics of eszopiclone for UV measurements on page 8.

 Calibration table

Abs at λmax 304 Concentration µg/ml

0 0

0.197 6

0.281 9

0.359 12

0.447 15

0.521 18

0.607 21

0.696 24

0.764 27

0.859 30

0.934 33

Figure 4 : calibration curve of eszopiclone

This figure was added to line 146 page 8 and highlighted.

3. Were ethics obtained to test the taste of films in volunteers and what were exclusion and inclusion criteria?

Response: 

The affirmed protocol set by the ethics committee of the Faculty of Pharmacy, Cairo University, Egypt with serial number: PI (2313) and the date of this approval (26 / 11 /2018).

Inclusion criteria:

• Human Subjects from 18 to 55 years of age (inclusive of both).

• Subjects within the BMI range from 18 to 35 kg/m2.

• Subjects with normal range of vital signs (Blood Pressure, Pulse Rate, Respiratory Rate and Body Temperature).

• Subjects with normal Medical and Surgical history without illness within the last 4 weeks prior to start of the study.

• Subjects with normal functioning of Cardiovascular, Respiratory, Gastrointestinal, Nervous System, Musculoskeletal, Vascular, GenitoUrinary, Endocrine/Metabolic systems.

• Subjects with normal Lymph nodes, head, neck, eyes, ears, nose, throat and skin. 

• Subjects with normal laboratory investigations

• Subjects able to communicate effectively.

• Subjects willing to provide informed consent and adhere to the protocol requirements.

• Urine analysis for narcotic drug (for all subjects) / Pregnancy test (for females) will be conducted prior to each phase.

Exclusion Criteria:

• Contraindications or Hypersensitivity to Eszopiclone or related group of drugs. 

• History or presence of any medical condition or disease according to the opinion of the physician.

• History or presence of significant alcoholism or drug abuse in the past one year.

• History or presence of significant smoking (more than 09 cigarettes/day or consumption of tobacco products and refusal to restrain from smoking or consumption of tobacco products for 48.00 hours before dosing until checkout).

• History or presence of significant renal or hepatobiliary problems.

• History or presence of significant asthma, urticaria or other allergic reactions.

• History or presence of significant gastric and/or duodenal ulcers.

• Difficulty in donating blood.

• Difficulty in swallowing film coated tablet or capsules. 

• Systolic blood pressure less than 90 mm Hg or more than 140 mm Hg. 

• Diastolic blood pressure less than 60 mm Hg or more than 100 mm Hg.

• Pulse rate less than 50/minute or more than 100/minute.

• Use of any prescribed medication during last two weeks or OTC medicines or medicinal products during the last one week preceding the first dosing.

• Major illness during 3 months before screening.

• Subjects who have been on an abnormal diet during the four weeks before screening.

• Participation in a drug research study within past 3 months.

• Donation of blood in the past 3 months before screening.

• Refusal to abstain from water for at least one hour prior to study drug administration and for at least one hour post dose.

• Refusal to abstain from food for at least ten hours before dosing and for at least 3 - 4 hours’ post dose.

• Refusal to abstain from alcohol or methylxanthine-containing beverages or foods (coffee, tea, carbonated drinks, chocolate) from 2 days prior to dosing till last sample collection.

• HIV positive.

• Anti-HCV antibodies positive.

• Found positive in the Urine drugs of abuse done at the time of check-in at the beginning of each period.

• Evidence of an uncooperative attitude.

4. It is claimed on page 11, Line 225 that the LC-MS data were validated, where is the data or reference? What were the limit of detection and limit of quantification?

Response: 

LC-MS were validated as seen in the calibration curve:

conc (ng/ml) peak area of IS peak area of drug PAR

 413949 0 0

0.5 418411 29767 0.07114297

5 456679 36747 0.08046571

10 408909 66537 0.16271836

15 437259 108534 0.24821444

25 497676 169853 0.34129233

50 489784 263185 0.53734912

75 401758 405673 1.00974467

100 441067 653450 1.48152095

The limit of detection is 0.05 ng/ml and the limit of quantification is 0.5 ng/ml.

5. Please use past tense for line 257.

Response: past tense was used and corrected in the revised manuscript. 

6. Line 284, please cite a reference.

Response: the reference was added to line 284.

7. How did the volunteers hold a film for 10 minutes in the mouth, did they chew them and then swallow?

Response: our formula was prepared as fast dissolving film to achieve higher plasma concentration in lower tmax and also to decrease residence time in the mouth to enhance patient compliance for this very bitter drug. The disintegration time of all formula were found to be between 10 to 40 seconds. A written mistake of the time was correct to sec. The correct time has been written in the revised manuscript in Table 2 (In vitro evaluation of medicated fast dissolving films (FDF)), page 14.

8. Did the authors obtain GMP grades of active ingredient and excipients to prepare the oral films for volunteers? Who supplied these?

Response:

In the material section point 2.1 line 76 page 4. Chemicals and Excipients we described where we obtained each material and each chemical used was supplied with material safety data sheet (MSDS).

9. A statistical method is described in the Method section but the results are not presented.

Response:

All statistical results are attached with the revised manuscript. 

A one-way analysis of variance (ANOVA) followed by the least significant difference (LSD) as a post hoc test was applied; using SPSS program version 17 software. The differences were considered significant if P<0.05. The column effluent was detected spectrophotometrically at 304 nm. Retention time for aripiprazole was 1.929 min which is well differentiated from the peak of the eszopiclone drug which has retention time 1.865 min as shown in figure (3b) which also revealed that there were no peaks due to formula compounds that might interfere with the assay. Calibration curve were constructed for eszopiclone with an equation that best describe the curve is (y = 0.0137 x – 0.0102 ) as y is represented by peak area and x for the concentration in (ng/ml) (r2 = 0.9774 ) as shown in figure (3d).The limit of detection and the limit of quantitation were 0.05 and 0.5 (ng/ml) respectively.

Discussion: When we compare the reference tablet with the three fast dissolving films containing sodium alginate or xanthan gum or pullulan as polymer to study its effect on Cmax , AUC 0-24 and AUC 0-INF. A significant difference (p-value < 0.05) was found between the reference and F4 which contain sodium alginate as polymer in all parameters tested as seen in the sheet attached with our response. Tmax is decreased in F4 formulae which contain sodium alginate as polymer and it was 1.063 h for F4 while 2.25 h only for commercial tablet . Relative bioavailability is increased 1.6 fold for F4, 1.2 fold for F2 and 1.1 fold for F6.

1. Ponrasu, T., et al., Fast Dissolving Electrospun Nanofibers Fabricated from Jelly Fig Polysaccharide/Pullulan for Drug Delivery Applications. Polymers, 2021. 13(2): p. 241.

2. Abu-Huwaij, R., et al., Formulation and in vitro evaluation of xanthan gum or carbopol 934-based mucoadhesive patches, loaded with nicotine. AAPS PharmSciTech, 2011. 12(1): p. 21-27.

3. Phaechamud, T. and G.C. Ritthidej, Formulation variables influencing drug release from layered matrix system comprising chitosan and xanthan gum. AAPS PharmSciTech, 2008. 9(3): p. 870-877.

4. Senturk Parreidt, T., K. Müller, and M. Schmid, Alginate-Based Edible Films and Coatings for Food Packaging Applications. Foods (Basel, Switzerland), 2018. 7(10): p. 170.

5. Duckworth, P.F., et al., Alginate films augmented with chlorhexidine hexametaphosphate particles provide sustained antimicrobial properties for application in wound care. Journal of materials science. Materials in medicine, 2020. 31(3): p. 33-33.

6. Zulkiflee, I. and M.B. Fauzi, Gelatin-Polyvinyl Alcohol Film for Tissue Engineering: A Concise Review. Biomedicines, 2021. 9(8): p. 979.

7. Karthikeyan, D., S. Sri, and C.S. Kumar, Development of Fast Dissolving Oral Film Containing of Rizatriptan Benzoate as an Anti Migraine Medication. Indo American Journal of Pharmaceutical Research, 2013. 3(3): p. 2642-2654.

---

## [Decision Letter · Decision Letter 1]

14 Mar 2022

Construction of Sublingual Trilaminated Eszopiclone Fast Dissolving Film for Treatment of Insomnia: Formulation, Characterization and In-vivo Clinical Comparative Pharmacokinetic Study in Healthy Human Subjects.

PONE-D-21-37028R1

Dear Dr. Teaima,

We’re pleased to inform you that your manuscript has been judged scientifically suitable for publication and will be formally accepted for publication once it meets all outstanding technical requirements.

Kind regards,

José das Neves

Academic Editor

PLOS ONE

Additional Editor Comments (optional):

Please address the comment of Reviewer #1 regarding language during proofing of the article.

Reviewers' comments:

Reviewer's Responses to Questions

**Comments to the Author**

1. If the authors have adequately addressed your comments raised in a previous round of review and you feel that this manuscript is now acceptable for publication, you may indicate that here to bypass the “Comments to the Author” section, enter your conflict of interest statement in the “Confidential to Editor” section, and submit your "Accept" recommendation.

Reviewer #1: All comments have been addressed

Reviewer #2: All comments have been addressed

2. Is the manuscript technically sound, and do the data support the conclusions?

Reviewer #1: Yes

Reviewer #2: Yes

3. Has the statistical analysis been performed appropriately and rigorously? 

Reviewer #1: Yes

Reviewer #2: Yes

4. Have the authors made all data underlying the findings in their manuscript fully available?

Reviewer #1: Yes

Reviewer #2: Yes

5. Is the manuscript presented in an intelligible fashion and written in standard English?

Reviewer #1: (No Response)

Reviewer #2: Yes

6. Review Comments to the Author

Reviewer #1: Comments for ‘Construction of Sublingual Trilaminated Eszopiclone Fast Dissolving Film for

Treatment of Insomnia: Formulation, Characterization and In-vivo Clinical Comparative

Pharmacokinetic Study in Healthy Human Subjects.’

Ref. No. PONE-D-21-37028R

I thank the editor for giving me an opportunity to review this manuscript. All major corrections are done but still language is required to improve.

Eg pg 16 of manuscript, line 318 -So it is Require special packaging for products stability and safety’ this can be written as Special packaging is required for stability and safety of product.

Please note, paper can be accepted after minor /grammatical correction, no need to review again.

Reviewer #2: Dear authours many thanks for addrssing the points and making alterations in the manuscript. It is an interesting study.

7. PLOS authors have the option to publish the peer review history of their article (what does this mean?). If published, this will include your full peer review and any attached files.

Reviewer #1: No

Reviewer #2: **Yes: **Dr Touraj Ehtezazi, Reader in Pharmaceutics

---

## [Editor Report · Acceptance letter]

17 Mar 2022

PONE-D-21-37028R1 

Construction of Sublingual Trilaminated Eszopiclone Fast Dissolving Film for the Treatment of Insomnia: Formulation, Characterization and *In vivo* Clinical Comparative Pharmacokinetic Study in Healthy Human Subjects 

Dear Dr. Teaima:

I'm pleased to inform you that your manuscript has been deemed suitable for publication in PLOS ONE. Congratulations! Your manuscript is now with our production department. 

Kind regards, 

on behalf of

Dr. José das Neves 

Academic Editor

PLOS ONE